# Protective Effects of Quercetin on Oxidative Stress-Induced Tubular Epithelial Damage in the Experimental Rat Hyperoxaluria Model

**DOI:** 10.3390/medicina57060566

**Published:** 2021-06-03

**Authors:** Ahmet Guzel, Sedat Yunusoglu, Mustafa Calapoglu, Ibrahim Aydın Candan, Ibrahim Onaran, Meral Oncu, Osman Ergun, Taylan Oksay

**Affiliations:** 1Department of Urology, Aydın State Hospital, Aydın 09100, Turkey; 2Department of Urology, Afyonkarahisar State Hospital, Afyonkarahisar 03100, Turkey; sedat.yunusoglu@saglik.gov.tr; 3Department of Biochemistry, Faculty of Arts and Science, Suleyman Demirel University, Isparta 32100, Turkey; calapoglu@hotmail.com; 4Department of Histology and Embryology, Faculty of Medicine, Alanya Alaaddin Keykubat University, Antalya 07100, Turkey; ibrahim.candan@alanya.edu.tr; 5Department of Medical Biology, Faculty of Medicine, Suleyman Demirel University, Isparta 32100, Turkey; ibrahimonaran@sdu.edu.tr; 6Department of Histology and Embryology, Faculty of Medicine, Suleyman Demirel University, Isparta 32100, Turkey; meraloncu@sdu.edu.tr; 7Department of Urology, Faculty of Medicine, Suleyman Demirel University, Isparta 32100, Turkey; osmanergun@sdu.edu.tr (O.E.); taylanoksay@sdu.edu.tr (T.O.)

**Keywords:** calcium oxalate, hyperoxaluria, oxidative stress, quercetin

## Abstract

*Background and Objectives*: The most common kidney stones are calcium stones and calcium oxalate (CaOx) stones are the most common type of calcium stones. Hyperoxaluria is an essential risk factor for the formation of these stones. Quercetin is a polyphenol with antioxidant, anti-inflammatory, and many other physiological effects. The aim of this study was to investigate the protective effect of quercetin in hyperoxaluria-induced nephrolithiasis. *Materials and Methods*: Male Wistar-Albino rats weighing 250–300 g (*n* = 24) were randomized into three groups: Control (*n* = 8), ethylene glycol (EG) (*n* = 8), and EG + quercetin (*n* = 8). One percent EG-water solution was given to all rats except for the control group as drinking water for five weeks. Quercetin-water solution was given to the EG + quercetin group by oral gavage at a dose of 10 mg/kg/day. Malondialdehyde (MDA), catalase (CAT), urea, calcium, and oxalate levels were analyzed in blood and urine samples. Histopathological assessments and immunohistochemical analyses for oxidative stress and inflammation indicators p38 mitogen-activated protein kinase (p38-MAPK) and nuclear factor kappa B (NF-kB) were performed on renal tissues. *Results*: The MDA levels were significantly lower in the quercetin-treated group than in the EG-treated group (*p* = 0.001). Although CAT levels were higher in the quercetin-treated group than the EG-administered group, they were not significantly different between these groups. The expression of p38 MAPK was significantly less in the quercetin-treated group than the EG group (*p* < 0.004). There was no statistically significant difference between the quercetin and EG groups in terms of NF-kB expression. *Conclusions*: We conclude that hyperoxaluria activated the signaling pathways, which facilitate the oxidative processes leading to oxalate stone formation in the kidneys. Our findings indicated that quercetin reduced damage due to hyperoxaluria. These results imply that quercetin can be considered a therapeutic agent for decreasing oxalate stone formation, especially in patients with recurrent stones due to hyperoxaluria.

## 1. Introduction

Urinary stone formation involves several factors, including the concentration of ions causing stone formation, urine pH, urinary flow rates, and urinary tract anatomy [1]. Approximately 75% of the renal stones are calcium stones, and calcium oxalate (CaOx) stones are the most commonly encountered calcium stones [2,3]. It is known that hyperoxaluria is an essential risk factor for the formation of these stones. The resultant increase in urinary oxalate concentration induces stone formation via calcium and oxalate ions to form CaOx crystals at physiological pH. It was demonstrated that the excess oxalate concentration damages kidney tubule cells via free oxygen radicals and increased lipid peroxidation and this process facilitates both the formation of CaOx crystals and their attachment to the kidney tubular epithelium [4,5].

Despite the improvements in renal stone treatment, the recurrence rates of 10% in one year, 35% in five years, and 50% in ten years were reported for calcium oxalate stones [6,7]. These high recurrence rates have led researchers to explore potential preventive methods for the recurrence of CaOx stones [6,7,8].

Quercetin is a flavanol found in various foods, including nuts, wine, seeds, and some fruits and vegetables [8]. In addition to its potent antioxidant, radical scavenging, and anti-inflammatory activities, it also has antibacterial, antiviral, gastroprotective, and immune-modulating activities [8,9,10].

An increase in oxidative stress was shown in patients with kidney stones [11]. As such, a lower risk of nephrolithiasis was reported in patients with high antioxidant levels than those with low antioxidant levels [12]. It was shown that the Randall plaques, which are thought to be formed due to oxidative stress, have an essential role in CaOx stone recurrence [13]. Therefore, inhibition of Randall plaque formation and oxidative stress may be beneficial for preventing stone recurrence.

This study investigated the possible protective effect of quercetin in nephrolithiasis and nephrocalcinosis by determining its impact on oxidative stress and inflammation in a rat hyperoxaluria model.

## 2. Materials and Methods

### 2.1. Animals

Male Wistar-Albino rats weighing 250–300 g (*n* = 24) were used in this experimental study, conducted at the Süleyman Demirel University Experimental Animal Laboratory. Rats were housed in a room maintained at 25 ± 1 °C with 55% relative humidity. They were given food and water ad libitum. The study was conducted after taking approval from Süleyman Demirel University Animal Experiments Ethical Review Committee (approval number: 33/01). All experiments were performed based on the ethical principles reported in European Union Directive 2010/63/EU for animal experiments.

### 2.2. Chemicals

Ethylene glycol (EG) was purchased from Merck Chemicals (Darmstadt, Germany), while quercetin was purchased from Sigma Chemicals (St. Louis, MO, USA).

### 2.3. Hyperoxaluria-Induced Nephrolithiasis Model and Quercetin Administration

The rats were randomly divided into three groups; control (*n* = 8), EG (*n* = 8), and EG + quercetin (*n* = 8) groups. All rats except for the control group were given 1% EG–water solution as drinking water for five weeks [14]. A 10 mg/kg/day dose of quercetin-water solution was administered to the rats in the EG + quercetin group by oral gavage. Twenty-four-hour urine samples were collected from the animals on day 0 before administering EG or quercetin and on days 15 and 30. Rats were anesthetized at the end of the five-week experiment period, and blood samples were collected from the inferior caval vein. Then, the rats were sacrificed, and both kidneys were rapidly excised. Right kidneys were embedded in 10% neutral-buffered formaldehyde. Left kidneys were washed with cold 0.9% sodium chloride solution and stored at −80 °C until homogenization.

### 2.4. Biochemical Analyses

Analysis of the plasma oxalate levels was performed by the technique previously described by Ladwig et al. [15]. Urinary oxalate level was determined by the enzymatic Trinity Biotech Oxalate kit (Trinity Biotech pic, St. Louis, MO, USA). Catalase (CAT) is an antioxidant enzyme [10,16]. Catalase activity was measured by the Aebi method [16]. Supernatant protein levels were analyzed by bovine serum albumin using the Lowry method as described in the standard protocol [17]. Malondialdehyde (MDA) levels were also analyzed since MDA is considered a biomarker of oxidative stress [9]. Malondialdehyde levels were measured spectrophotometrically as per the method proposed by Ohkawa et al. [18]. Protein detection was performed using the method described by Lowry et al. [17]. The creatinine levels of plasma and urine were measured by colorimetry as per the Jaffe method; plasma urea concentrations were analyzed by kinetic-colorimetric methods and urease and glutamate dehydrogenase reactions. In contrast, urine calcium levels were measured by an autoanalyzer via Schwarzenbach and the o-Cresolphthalein complex method. Timed urine and blood samples were collected for calculating creatinine clearance. Urine volume (V) was expressed in mL, while serum and urine creatinine levels were expressed in mg/dl. Creatinine clearance was calculated according to the formula:Creatinine clearance (mL/min) = U (mg/dL) × V (mL/min)/S (mg/dL) [V: Urine volume (mL/min), U: Urine creatinine (mg/dL), S: Serum creatinine (mg/dL)]

### 2.5. Histopathological Analyses

The right kidneys were embedded into 10% neutral-buffered formaldehyde (MerckKGaA, Darmstadt, Germany). The tissue processing was initiated after a 24–48 h fixation period. Kidney tissues were stained with hematoxylin and eosin (H&E) and examined under a light microscope. Histopathological examinations were performed by 10×, 20×, and 40× magnifications, particularly for assessing vascular congestion in both medulla and cortical glomeruli, the intensity of mononuclear cell infiltration, and sizes and shapes of the tubular lumina.

### 2.6. Immunohistochemical Analyses

Nuclear transcription factor-kB (NF-kB) and p38 mitogen-activated protein kinase (p38-MAPK) are the two main signaling pathways at the protein level in proinflammatory cytokine biosynthesis [19]. In addition, oxalate selectively activates the p38-MAPK signaling pathway [20]. Slides were taken into the incubation container after blockage. The NF-kB/p65 antibody (Santa Cruz, sc-109) was added to the first section, MAPK/p38 antibody (Santa Cruz, sc-7149) to the second section, and a secondary antibody (Goat anti-polyvalent, Thermo) was added to the subsequent sections for control purposes and kept at +4 °C overnight. The ABC Staining Kit (Santa Cruz, sc-2018) was used for the rest of the staining protocol. The tissue covered with lamella using Entella was stored and dried for examining with a light microscope. The sections were graded as follows, depending on the intensity of staining in light microscopy, as in the study by Liu HS et al. [19]:

(−) score (negative score): no staining;

(+) score (1 positive score): mild;

(++) score (2 positive scores): moderate;

(+++) score (3 positive scores): complete staining is present.

### 2.7. Statistics

Statistical analysis was performed using Statistical Package for Social Sciences 11.0 (SPSS 11.0) software. Values were given as means ± standard deviations. The normal distribution of the variables was tested by a single sample Kolmogorov-Smirnov test. A one-way ANOVA was conducted and, if shown to be significant, Duncan’s post hoc test was used to evaluate and determine the differences between the groups. The *p* value was considered significant when it was below <0.05.

## 3. Results

### 3.1. Biochemical Results

Study group data are shown in Table 1. The mean MDA level of the EG group was higher than the control group, while the mean MDA level of the quercetin group was significantly lower than the EG group (*p* = 0.001). There was no statistically significant difference between the groups in terms of CAT activity. In our study, MDA levels were reduced by quercetin, while CAT activity was not affected by quercetin. Urinary oxalate levels were significantly higher in the EG group than the other groups (*p* < 0.001). However, the urinary oxalate levels of the quercetin group were significantly lower than the EG group (*p* < 0.001). These results indicated that quercetin reduced the urinary oxalate levels, which were increased by EG administration. While the plasma urea concentration increased significantly in the EG group compared to in the control group, a statistically significant decrease was observed in the quercetin group compared to the EG group (*p* < 0.05). There was no statistically significant difference between the groups in terms of creatinine clearance and plasma oxalate levels (*p* > 0.05). Although quercetin decreased plasma oxalate level, this was not statistically significant. Urinary calcium concentration was significantly lower in the other groups than the control group (*p* < 0.05). However, there was no statistically significant difference between EG and quercetin groups in terms of mean urinary calcium concentrations. Altogether these findings indicate that EG increased urinary oxalate excretion and plasma urea concentration and decreased urinary calcium concentrations.

### 3.2. Histopathological Results

Histopathological examinations of the control group rat kidneys under light microscopy revealed normal findings (Figure 1-Control). Findings, including congestion in glomerular vessels, mononuclear cell infiltration, and narrowing of the tubular lumens, especially in the proximal tubules, were detected in the EG group (Figure 1-EG). Similar histopathological findings were observed in the quercetin group, but they were all milder (Figure 1-EG + quercetin). However, no scoring system was used to evaluate whether this difference was statistically significant.

### 3.3. Immunohistochemical Results

The impact of quercetin on p38 mitogen-activated protein kinase (p38-MAPK) and nuclear factor kappa B (NF-kB) was investigated by immunohistochemical staining of the renal tissues (Figure 2). Comparison of the groups in terms of p38-MAPK elucidated that renal tissues of the EG group rats were stained significantly more intensely than those of the control group rats (*p* < 0.001). On the other hand, p38-MAPK staining was significantly less intense in the quercetin group than in the EG group. The statistical analysis revealed that quercetin reduced p38-MAPK activity significantly compared to the EG group (*p* < 0.004).

A comparison of the groups concerning NF-kB immunostaining revealed that renal tissues of the rats in the EG group were stained significantly more intensely than those of the rats in the control group (*p* < 0.001). There was no statistically significant difference between the other groups in terms of the intensity of NF-kB staining. Altogether, these findings pointed out that EG induced the p38-MAPK and NF-kB activities in renal tissues. Additionally, quercetin significantly reduced p38-MAPK activity while it had no significant effect on NF-kB activity. The relevant data are displayed in Table 2.

## 4. Discussion

Despite advances in minimally invasive methods for treating nephrolithiasis, high residual stone and recurrence rates still constitute severe problems for the patients and urologists [21]. Therefore, effective medical treatment strategies are needed.

The exact mechanisms of stone formation have not been identified; however, it is known that approximately 80% of kidney stones contain CaOx and one of the most common causes of CaOx stones is hyperoxaluria [1,22]. Oxalate is a by-product of normal metabolism, eliminated from the body under normal conditions [20]. In hyperoxaluria, excess urinary oxalate combines with calcium at physiological pH and forms CaOx crystals, accumulating in the kidneys. The CaOx crystals were shown to damage kidney tubular epithelial cells and lead to renal stone disease [14,20]. Therefore, hyperoxaluria rat models were widely used to mimic kidney stone formation in humans. In these studies, EG was used as a hyperoxaluria-inducing agent, as it was the case in our study [14,20].

It was reported that the oxalate concentration in urine increased within two days, hyperoxaluria developed within three days, CaOx crystalluria developed within two weeks, and CaOx nephrolithiasis was observed within 4–6 weeks when 0.75–1% EG was administered for inducing hyperoxaluria [14,22]. At the end of this process, other urinary factors and creatinine clearance remained within normal limits, while urinary pH and citrate excretion were significantly reduced [14,22]. In our study, we induced hyperoxaluria by administering 1% EG, which subsequently led to cellular damage in tubular cells.

In hyperoxaluria, accumulation of the CaOx crystals in renal papilla collector duct lumina and resultant stone-like deposits lead to the obstruction, which paves the way for renal deterioration. As a result of this process, blood levels of nitrogenous waste products, such as uric acid, blood urea nitrogen, and creatinine, may increase [22,23]. In our study, renal dysfunction was demonstrated by a marked increase in serum urea levels in the animals treated with EG. However, quercetin treatment significantly reduced serum urea levels.

Quercetin is a molecule with potent antioxidant and anti-inflammatory effects [24]. It is found in vegetables, fruits, tea, and numerous types of food. In addition to being the strong scavenger of reactive oxygen species (ROS), quercetin increases the total plasma antioxidant capacity [25,26]. It was reported in some studies that flavonoids reduce oxidative stress in kidney tubular cells, prevent the accumulation of CaOx crystals, and reduce lipid peroxidation induced by oxalate in cell cultures [27,28].

It is widely accepted that oxidative stress plays a significant role in the formation of kidney stones [12,22]. The CaOx crystals accumulated in the kidney tissue leading to the synthesis of several macromolecules that initiate inflammatory and fibrogenic processes. Reactive oxygen species are most likely involved in various signaling events during this period [14]. In both animal and kidney epithelial cell culture studies, the free radical formation was shown in response to hyperoxaluria and CaOx crystal formation [20,22]. Since lipids are the most sensitive biomolecules in the cell membrane against free radicals, lipid peroxidation occurs in the cell membranes [27,28]. Lipid peroxidation is the primary event of cellular damage since it disrupts the membrane structure and subsequently causes cell damage. The MDA is an indicator of lipid peroxidation; it also indicates that ROS are overproduced [22,29]. Therefore it is used in the evaluation of lipid peroxidation. It is known that the damage caused by CaOx crystal deposition and the formation of ROS can be prevented or reduced by endogenous antioxidants such as superoxide dismutase (SOD), CAT, and glutathione peroxidase (GSH-Px) [20,22,29]. In the present study, MDA levels were found to increase in an experimental animal model of hyperoxaluric nephrolithiasis, which was successfully created by ethylene glycol administration. Our study also showed that quercetin reduced the oxidative stress-enhancing effect of hyperoxaluria, as demonstrated by both biochemical and histopathological parameters.

It is known that ROS activate signal molecules such as protein kinase C (PKC), c-Jun N-terminal kinase (JNK), and p38-MAPK [21]. Activation of these signal molecules leads to the induction of NF-kB and active protein-1 (AP-1) transcription factors. As a result of these events, expression of the proteins such as monocyte chemoattractant protein-1 (MCP-1), osteopontin (OPN), fibronectin, and transforming growth factor beta 1 (TGF-β1) increase [20]. These proteins facilitate the adhesion of CaOx crystals and the relevant inflammatory processes [20]. Peerapen et al. reported that p38-MAPK signaling pathways mediated disruption of the tight connections between the epithelial cells [30]. They also showed that the expression of the proteins involved in the p38-MAPK signaling pathway increased during calcium oxalate stone formation. It was also demonstrated that oxalate selectively activated the p38-MAPK signaling pathway, which had an essential role in the nephrotoxicity of oxalate in human kidney epithelial cells [31]. Shiyong Qi et al. proved that CaOx crystals induced the enhancement of adhesion molecules in proximal tubular epithelial cells, and this process was mediated by the p38-MAPK signaling pathway [31]. Our study showed that oxalate induced p38-MAPK and NF-kB activation. Additionally, we showed that quercetin significantly reduced the activity of p38-MAPK. Based on this information, it can be suggested that quercetin can prevent the recurrence of CaOx stones.

The study has some limitations. The first is that there was no control group administered quercetin alone. Therefore, the effect of orally administered quercetin on the production and intestinal absorption of oxalate has not been clearly evaluated. Second, the effect of quercetin on ethylene glycol absorption was also not evaluated. Consequently, further research is needed to clarify the mechanism of action of quercetin on hyperoxaluria.

## 5. Conclusions

We conclude that quercetin inhibited the inflammatory and oxidative processes triggered by hyperoxaluria in renal tissues. Since these processes lead to CaOx stone formation, quercetin can be considered in the preventive medical treatment of recurrent CaOx stones. Nevertheless, further experimental and clinical research is needed to validate our findings.

## Figures and Tables

**Figure 1 medicina-57-00566-f001:**
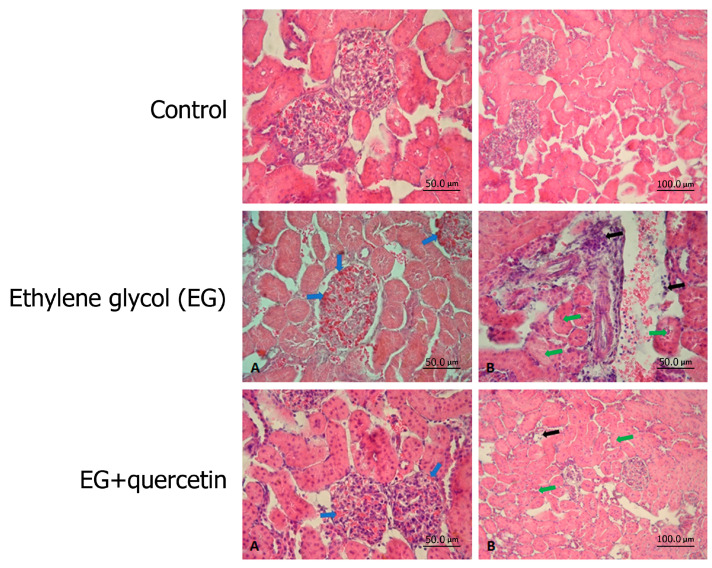
Hematoxylin and eosin (H&E) stain of kidney tissue and histopathological assessment of all groups. Control: Samples with normal kidney tissue structure. Ethylene glycol (EG): (B) Sample with apparent mononuclear cell infiltration in kidney medulla and cortex layers (black arrow), as well as narrowing of the lumen of the tubules (green arrow). (A) Sample with apparent vascular congestion in both medulla and cortical glomeruli (blue arrow). EG + quercetin: (A) In this tissue sample, decreased vascular congestion in both the medulla and cortical glomeruli are observed. (blue arrow). (B) In this tissue sample, decreased mononuclear cell infiltration in renal medulla and cortex layer (black arrow), and a return to normal structure in the lumen of tubules (green arrow) are observed.

**Figure 2 medicina-57-00566-f002:**
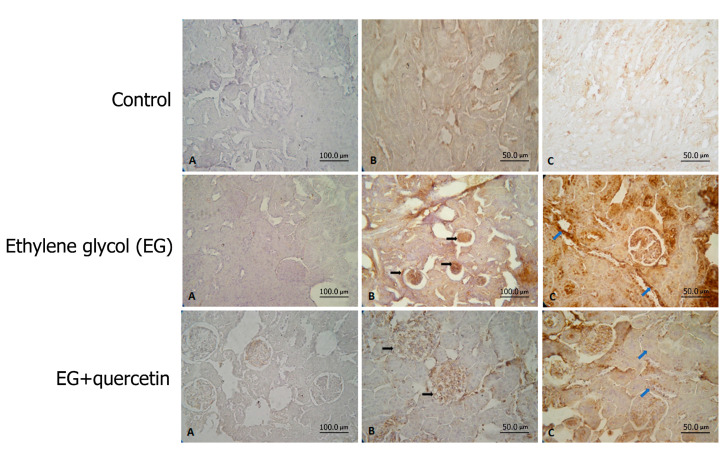
p38-MAPK and NF-kB stain of kidney tissue, and immunohistochemical assessment of all groups. Control: Samples with normal kidney tissue structure. Positive staining of p38-MAPK and NF-kB is not detected in the samples. Staining control (A), p38-MAPK (B) and NF-kB (C). Ethylene glycol (EG): Significant positive staining of p38-MAPK and NF-kB is detected in the samples compared to the control group. Staining control (A), p38-MAPK (B) and NF-kB (C). EG + quercetin: Decreased positive staining of p38-MAPK and NF-kB is detected in the samples compared to the ethylene glycol group. Staining control (A), p38-MAPK (B) and NF-kB (C).

**Table 1 medicina-57-00566-t001:** Statistical data of the study groups and Duncan’s multiple range test results.

Biochemical Variables	Control (*n* = 8)	EG (*n* = 8)	EG + Quercetin (*n* = 8)	*p* Values
MDA (nmol/mg protein)	4.53 ± 0.89 ^a^	7.40 ± 1.64 ^b^	5.41 ± 0.82 ^a^	0.0001
CAT (µmol/min/mg)	118.41 ± 26.02 ^a^	119.48 ± 16.55 ^a^	131.06 ± 28.99 ^a^	0.360
P_Urea_ (mg/dL)	42.37 ± 3.37 ^a^	86.75 ± 61.44 ^b^	51.62 ± 19.97 ^a^	0.040
U_Calcium_ (mg/dL)	12.90 ± 3.37 ^a^	8.38 ± 3.39 ^b^	9.58 ± 5.59 ^b^	0.046
Creatinine Clearance (mL/min)	0.26 ± 0.09 ^a^	0.50 ± 0.28 ^a^	0.42 ± 0.25 ^a^	0.174
P_Oxalate_ (µmol/L)	32.18 ± 19.28 ^a^	71.92 ± 61.83 ^a^	47.48 ± 20.09 ^a^	0.117
U_Oxalate_ (µmol/24 h)	2.72 ± 0.36 ^a^	10.14 ± 4.13 ^b^	6.81 ± 2.26 ^c^	0.0001

Data shown in different letters such as ^a, b, c^ denotes *p* < 0.05. EG ethylene glycol, MDA malondialdehyde, CAT catalase, P_Urea_ plasma urea levels, U_Calcium_ urine calcium levels, P_Oxalate_ plasma oxalate levels, U_Oxalate_ urine oxalate levels.

**Table 2 medicina-57-00566-t002:** Statistical data between groups in terms of p38-MAPK and NF-kB (Mann-Whitney test).

Immunohistochemical Variables	Comparison of Groups Control and EG	Comparison of Groups Control and EG + Quercetin	Comparison of Groups EG and EG + Quercetin
p38-MAPK	*p* < 0.001	*p* > 0.036	*p* < 0.004
NF-kB	*p* < 0.001	*p* > 0.227	*p* > 0.054

EG ethylene glycol, p38-MAPK p38 mitogen-activated protein kinase, NF-kB nuclear factor kappa B.

## Data Availability

The data of this study are available from the corresponding author upon reasonable request.

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
