# Peer review of "Protective Effects of Quercetin on Oxidative Stress-Induced Tubular Epithelial Damage in the Experimental Rat Hyperoxaluria Model"

_medicina, 2021, doi:10.3390/medicina57060566_

Round 1
Reviewer 1 Report
The authors show that the polyphenol, quercetin, could suppress much of the inflammation and oxidative stress associated with oral ingestion of ethylene glycol in rats. This data suggests that oral ingestion of quercetin by calcium oxalate kidney stone formers could alleviate their stone formation. The results were convincing except for the following points. 1) the method used to quantitate the immunochemical staining was not stated. 2) The plasma oxalate level is extremely high even in controls (Harris et al; J Lab Clin Med 2004; 144;45) indicating that the method tested only in humans) used was not suitable for rats. This measurement can be safely be removed as the urine value appears consistent with literature values. Minor points: Table 1 should read "creatinine" not creatine. In Conclusions should read "We conclude that quercetin...".
Author Response
22/05/2021
Dear Editor-in-Chief,
Thank you for giving me the opportunity to submit a revised draft of my manuscript titled "Protective Effects of Quercetin on Oxidative Stress-Induced Tubular Epithelial Damage in the Experimental Rat Hyperoxaluria Model." to Medicina. We appreciate the time and effort that you and the reviewers have dedicated to providing your valuable feedback on my manuscript. We are grateful to the reviewers for their insightful comments on my paper. We have been able to incorporate changes to reflect most of the suggestions provided by the reviewers. We have highlighted the changes within the manuscript.
Here is a point-by-point response to the reviewers’ comments and concerns.
Comments from Reviewer 1
The authors show that the polyphenol, quercetin, could suppress much of the inflammation and oxidative stress associated with oral ingestion of ethylene glycol in rats. This data suggests that oral ingestion of quercetin by calcium oxalate kidney stone formers could alleviate their stone formation. The results were convincing except for the following points.
- Comment: The method used to quantitate the immunochemical staining was not stated.
Response: Thank you for pointing this out. We agree with this comment. For this reason, we added a paragraph about the method used to quantitate the immunochemical staining in the Immunohistochemical analyses section of Materials and Methods as follows. In addition, we added a reference.
“The sections were graded as follows, depending on the intensity of staining in light microscopy, as in the study by Liu HS et al.
(-) score (negative score): no staining.
(+) score (1 positive score): mild.
(++) score (2 positive scores): moderate.
(+++) score (3 positive scores): Complete staining is present.”
- Comment: The plasma oxalate level is extremely high even in controls (Harris et al; J Lab Clin Med 2004; 144;45) indicating that the method tested only in humans) used was not suitable for rats. This measurement can be safely be removed as the urine value appears consistent with literature values.
Response: You have made an important point here. Thank you for pointing this out. However, we used this method as there are studies that used the method for plasma oxalate determination we chose (Li Y, McMartin KE. Strain differences in urinary factors that promote calcium oxalate crystal formation in the kidneys of ethylene glycol-treated rats. Am J Physiol Renal Physiol. 2009 May; 296(5):F1080-7.). Since we want to present the results of all parameters analyzed in our study, we have shown the plasma oxalate levels in Table 1. The interests of the readership of the journals have a wide range distribution. Therefore, many readers who are related or unrelated to the topic can read this article. Oxalate level is one of the main data of our study. If urine oxalate level results are presented and plasma oxalate level results are not presented, questions may arise in the reader's mind such as; "Are the data in this study missing?" "Are some of the study's data kept secret and not presented?" For these reasons, although our plasma oxalate level results are statistically insignificant, we think that it would be more appropriate not to remove them.
- Comment: Minor points: Table 1 should read "creatinine" not creatine. In Conclusions should read "We conclude that quercetin...".
Response: Thank you for this suggestion. We revised the first sentence in the Conclusions and we corrected the word creatine in Table 1 to creatinine. In addition, we have highlighted the changes.
Additional clarifications
In addition to the above comments, the words that required minor spell check pointed out by the reviewers have been corrected by a native English speaker.
We look forward to hearing from you in due time regarding our submission and to respond to any further questions and comments you may have.
Sincerely,
Ahmet GÜZEL, M.D.
Address: Aydın State Hospital,
Department of Urology.
Hasan Efendi Mah. Kızılay Cad. No: 13,
09100, Aydin, Turkey.
Tel: +90 505 3039414
Mail: drahmetguzel@yahoo.com

Reviewer 2 Report
Guzel and coauthors have submitted an interesting manuscript suggesting that Quercetin may block kidney inflammation produced by oxaluria, thus reducing stone formation. The subject is of significance and novelty.
The reviewer suggests several modifications to strengthen the work
- It is notable that quercetin blunted the increase in both plasma and urine oxalate in the EG treated animals. Could the decrease in oxidative stress seen in the quercetin treated animals be simply due to a decrease in exposure to oxalate? The authors state in the text that there is no difference in plasma oxalate levels but they have "a" designation for all three values
- In that regard, it would be useful to have a control group of animals who had quercetin alone. Quercetin may have some underlying effect on oxalate production or intestinal absorption that plays an important role in the effect seen by the authors.
- The 24h urine calcium should be included.
- It would be interesting to include another marker of tubular damage such as NGAL or KIM1.
- The methodology for assessing and comparing the immunofluorescence intensity is not indicated.
- The reason for measuring MAPK and not other pathways associated with inflammation is not clear.
Author Response
22/05/2021
Dear Editor-in-Chief,
Thank you for giving me the opportunity to submit a revised draft of my manuscript titled "Protective Effects of Quercetin on Oxidative Stress-Induced Tubular Epithelial Damage in the Experimental Rat Hyperoxaluria Model." to Medicina. We appreciate the time and effort that you and the reviewers have dedicated to providing your valuable feedback on my manuscript. We are grateful to the reviewers for their insightful comments on my paper. We have been able to incorporate changes to reflect most of the suggestions provided by the reviewers. We have highlighted the changes within the manuscript.
Here is a point-by-point response to the reviewers’ comments and concerns.
Comments from Reviewer 2
Guzel and coauthors have submitted an interesting manuscript suggesting that Quercetin may block kidney inflammation produced by oxaluria, thus reducing stone formation. The subject is of significance and novelty.
The reviewer suggests several modifications to strengthen the work
- Comment: It is notable that quercetin blunted the increase in both plasma and urine oxalate in the EG treated animals. Could the decrease in oxidative stress seen in the quercetin treated animals be simply due to a decrease in exposure to oxalate? The authors state in the text that there is no difference in plasma oxalate levels but they have "a" designation for all three values
Response: You have made an important point here. Thank you for pointing this out. In Table 1, if there is no significant difference between the values, it is shown with the same letter, and if there is a significant difference between the values, it is shown with a different letter such as superscript a, b, c. Statistical evaluation shows that the decrease in plasma oxalate level is not significant. Therefore, quercetin may be said to reduce the increase in urinary oxalate in animals treated with EG, but it is not possible to say the same for plasma oxalate. For this reason, we think it is not possible to say that the decrease in oxidative stress seen in animals treated with Quercetin may be due to the decrease in oxalate exposure. That's why we added the following sentence to the results section to eliminate confusion and we have highlighted the changes.
“Although quercetin decreased plasma oxalate level, this was not statistically significant.”
- Comment: In that regard, it would be useful to have a control group of animals who had quercetin alone. Quercetin may have some underlying effect on oxalate production or intestinal absorption that plays an important role in the effect seen by the authors.
Response: Thank you for this suggestion. It would have been interesting to explore this aspect. In the literature review, we found studies showing the anti-inflammatory and antioxidant activity of quercetin, and we designed the study to evaluate the effectiveness of quercetin on inflammatory and oxidative stress and harmful effects caused by hyperoxaluria. Therefore, we did not create a control group that was administered quercetin alone to evaluate some possible effects of quercetin on oxalate production or intestinal absorption. However, your suggestion made a very valuable contribution for this study and we added a paragraph about study limitations en of the discussion as follows;
“The study has some limitations. The first is that there was no a control group administered quercetin alone. Therefore, the effect of orally administered quercetin on the production and intestinal absorption of oxalate has not been clearly evaluated. Second, the effect of quercetin on ethylene glycol absorption was also not evaluated. Consequently, further research is needed to clarify the mechanism of action of quercetin on hyperoxaluria.”
- Comment: The 24h urine calcium should be included.
Response: Thank you for pointing this out. The values in the row titled "Calcium (mg / dL)" shown in Table 1 are actually 24-hour urine calcium values. We mentioned in the materials and methods section as follows; "Twenty-four hour urine samples were collected from the animals on day 0 before administering EG or quercetin and on days 15 and 30." and "In contrast, urine calcium levels were measured by an autoanalyzer via Schwarzenbach and the o -Cresolphthaleincomplex method. " and as with oxalate, we did not use the UCalcium title as it lacked both plasma and urine values. However, in order to avoid confusion and to be clearly stated, we revised the title of "Calcium (mg / dL)" in Table 1 to "UCalcium (mg / dL)” and added its explanation to the table footnotes.
- Comment: It would be interesting to include another marker of tubular damage such as NGAL or KIM1.
Response: Thank you for this suggestion. It would have been interesting to explore this aspect. During study design and planning, NGAL was among the biomarkers we identified to examine inflammatory, oxidative stress, and tubular damage. However, we could not include this indicator in the study because the study budget was limited and insufficient. In addition, if I need to state with regret, it is not possible for us to conduct additional tests in our country due to some bureaucratic rules and lack of budget for the study.
- Comment: The methodology for assessing and comparing the immunofluorescence intensity is not indicated.
Response: Thank you for pointing this out. However, I guess there is a misunderstanding. Since immunofluorescent staining was not performed in our study, no evaluation was made related to this analysis. Immunohistochemical analysis was performed in our study, and we added the following a paragraph to the immunohistochemical analysis section of the materials and methods section to explain the assesment method.
“The sections were graded as follows, depending on the intensity of staining in light microscopy, as in the study by Liu HS et al.
(-) score (negative score): no staining.
(+) score (1 positive score): mild.
(++) score (2 positive scores): moderate.
(+++) score (3 positive scores): Complete staining is present.”
- Comment: The reason for measuring MAPK and not other pathways associated with inflammation is not clear.
Response: Thank you for pointing this out. We agree with this comment. Therefore, we have added the following sentence on p38 MAPK and NF-κB to the Immunohistochemical Analysis section of Materials and Methods.
"Nuclear transcription factor-κB (NF-κB) and p38 mitogen-activated protein kinase (p38 MAPK) are the two main signaling pathways at the protein level in proinflammatory cytokine biosynthesis. In addition, oxalate selectively activates the p38 MAPK signaling pathway."
In addition, in the last paragraph of the discussion section, we also mentioned cell studies that reported that the p38 MAPK signaling pathway selectively activated by oxalate mediates the deterioration of kidney epithelial cell connections and the formation of calcium oxalate stones. These studies also support the accuracy of our preference for p38 MAPK as an inflammation marker.
Additional clarifications
In addition to the above comments, the words that required minor spell check pointed out by the reviewers have been corrected by a native English speaker.
We look forward to hearing from you in due time regarding our submission and to respond to any further questions and comments you may have.
Sincerely,
Ahmet GÜZEL, M.D.
Address: Aydın State Hospital,
Department of Urology.
Hasan Efendi Mah. Kızılay Cad. No: 13,
09100, Aydin, Turkey.
Tel: +90 505 3039414
Mail: drahmetguzel@yahoo.com
